# Do Sex-Specific Factors Influence the Surgical Treatment of Facial Skin Cancer?

**DOI:** 10.3390/jpm13081193

**Published:** 2023-07-27

**Authors:** Sarah Victoria Wünscher, Stephan Spendel, Sebastian P. Nischwitz, Alessandro Gualdi, Alexander Avian, Lars-Peter Kamolz, Janos Cambiaso-Daniel

**Affiliations:** 1Division of Plastic, Aesthetic and Reconstructive Surgery, Department of Surgery, Medical University of Graz, 8036 Graz, Austria; sarah.wuenscher@stud.medunigraz.at (S.V.W.); sebastian.nischwitz@medunigraz.at (S.P.N.); lars.kamolz@medunigraz.at (L.-P.K.); janos.cambiaso-daniel@medunigraz.at (J.C.-D.); 2Department of Plastic, Reconstructive and Aesthetic Surgery, University Vita-Salute San Raffaele, 20132 Milan, Italy; alegualdi@yahoo.it; 3Milano Face Institute, 20146 Milan, Italy; 4Institute for Medical Informatics, Statistics and Documentation, Medical University of Graz, 8036 Graz, Austria; alexander.avian@medunigraz.at; 5COREMED-Cooperative Centre for Regenerative Medicine, JOANNEUM RESEARCH Forschungsgesellschaft mbH, 8036 Graz, Austria

**Keywords:** public health, sex ratio, surgery, skin neoplasms, incidence

## Abstract

Facial skin cancer (FSC) is prone to incomplete excision due to the sophisticated anatomy and the aesthetic importance of the face. In this study, we sought to investigate to what extent sex-specific differences and other operation-, patient-, and cancer-specific factors influence the re-resection rate in FSC surgery, in order to provide personalized treatment strategies to patients. In this retrospective study, patients (>18 years) undergoing surgical excision of an FSC were enrolled. Each patient’s demographic data, cancer location, the surgical team, primary and secondary surgeries were analyzed. Overall, 469 patients (819 surgeries) were included. The mean age was 69 ± 15 years. No significant association between sex-specific factors (surgeon’s sex (OR: 1.09, 95% CI: 0.76–1.56) or patient’s sex (OR: 0.85, 95% CI: 0.62–1.17), surgeon–patient sex concordance and discordance) and the likelihood of secondary surgery were found. However, healing by secondary intention (OR: 4.28; 95% CI: 1.94–9.45) and cancer location showed an increased re-resection rate. In conclusion, FSC surgery is a safe method unaffected by sex-specific factors, which had no impact on the re-resection rate. However, in further analysis, the likelihood of a re-resection was influenced by other factors such as healing by secondary intention and cancer location. This knowledge might be useful to provide an algorithm for personalized treatment strategies in the future.

## 1. Introduction

Skin cancer is the most common malign neoplasm in the human population and can be categorized into malignant melanoma (MM) and non-melanoma skin cancer (NMSC) [1]. NMSC represents most cases of skin cancer and can be further divided into cutaneous squamous cell carcinoma (cSCC) and basal cell carcinoma (BCC) [1]. Both tumor entities are characterized by an increasing incidence as well as a low mortality [2]. Basal cell carcinoma accounts for 80% of NMSC and is therefore described as the most common non-benign neoplasm of the human population [1,3]. Even though melanoma only represents 4% of skin cancer cases, it accounts for 65% of all skin-cancer related deaths due to its tendency for both lymphogenic and haematogenic metastasis [1,4].

Due to ultraviolet (UV) radiation being known as the main causative factor, sun exposed areas such as the face are important predilection sites for cancer development [1,3]. Owing to the rising incidence of skin cancer, the incidence of facial oncological surgery is likely to increase in the future [1,5,6,7]. Therefore, facial skin cancer (FSC) puts a huge economic burden on the health care system worldwide and is considered as a pre-eminent global public health problem [8,9].

Besides the magnificent economic impact, FSC poses an emotional, physical, and psychological strain on the patient [5]. The surgical excision of FSC, which is the treatment of choice in the majority of cases, can lead to visible scarring as well as extensive facial disfigurement [5,6]. Because of the centrality of the facial area to multiple aspects of life, anatomical or functional changes can result in social withdrawal, long absenteeism from work, and feelings of isolation [5].

To this day, surgical specialties are predominantly taken by men, despite the increasing number of female medical students [10]. Differences in the practice of medicine have been found between female and male physicians, where female physicians are more likely to use a patient-centered approach and adhere to clinical guidelines in comparison to their male colleagues [11]. A population-based cohort study published in JAMA *Surgery* showed that surgeon–patient sex discordance had a negative impact on patient outcomes [12].

To this day, to the authors’ knowledge, no study has investigated the impact of sex-specific differences as well as other operation-, patient-, and cancer-specific factors in the resection of FSC. Understanding which factors have an impact on the re-resection rate takes up great importance in optimizing the surgical treatment of FSC, by providing information in order to introduce an algorithm for a customized treatment. The aim of this study was to investigate to what extent sex-, patient-, operation-, and cancer-specific factors have an impact on the re-resection rate in FSC, in order to provide personalized treatment strategies for FSC patients.

## 2. Materials and Methods

We present a single-center retrospective study of 469 patients who underwent the excision of an FSC at the Division of Plastic, Reconstructive and Aesthetic Surgery at the Department of Surgery of the Medical University of Graz, Austria, from January 2005 to November 2021. The study was approved by the Institutional Review Board of the Medical University of Graz, Austria (EK 34-320 ex21/22).

### 2.1. Patients

All patients who underwent oncological surgery in a number of facial areas according to the mentioned location in the operation report (face, forehead, eye, lid, nose, cheek, mouth, lips, and chin) with the following ICD-10 diagnosis codes were enrolled: basal cell carcinoma (C44.0-4; BCC); cutaneous squamous cell carcinoma (C44.0-3; cSCC); malignant melanoma (C43.0-4; MM); and unspecified formations of the skin (D48.5), including eccrine carcinoma (EC) and melanoma in situ (MIS). Other types of skin cancers, benign lesions, and surgeries, which were documented as diagnostic incisional biopsies or tangential excisions, were not included in the study. In addition, all patients were diagnosed with a malign FSC before undergoing surgery. Only male and female patients older than 18 years were included in the study. After the initial automatic database collection, the medical files of all patients were evaluated individually. Patients who did not fit the inclusion criteria, those with metastatic skin cancer, and those who underwent surgery in a palliative setting were excluded from the study to avoid confounding effects. Figure 1 depicts the inclusion process. In a final step, all surgeries performed on these patients were screened.

### 2.2. Data Collection

The following data were collected manually from the computerized hospital database system for each surgery: type of anesthesia, surgery duration (in min), sex of the surgeon(s) (female/male), level of training of the surgeon(s), type of wound closure, resection safety margins (in mm), and if a re-resection was needed. To avoid confounding effects, surgery duration was only included in the statistical analysis if the wound was closed immediately after FSC excision either using direct closure or healing by secondary intention. Furthermore, cancer characteristics (type of cancer, cancer location, histological subtype of BCC) and patient factors (sex (female/male), age) were collected from the medical files of all patients.

Concerning the cancer location, the classification system of Aumüller et al. and Gualdi et al. were utilized with small modifications for further statistical analysis [13,14]. However, in order to obtain more powerful results, some areas—such as the frontal and temporal area, as well as the oral and mental area—were summarized. Figure 2 presents the used classification of FSC locations, being divided into the mouth, nasal, frontotemporal, periorbital, and buccal area.

Due to the different clinical appearance of the histopathological BCC subtypes, a sub-analysis of BCCs using the classification of Cameron et al. was performed [15]. If the BCC showed characteristics of more than one subtype according to the histopathological report, the subtype “mixed” was allocated to the cancer. Histopathological subtypes such as morpheaform clinically present as plaques with poorly defined borders, making preoperative markings challenging [15,16].

### 2.3. Statistical Analysis

Baseline characteristics are presented as mean ± SD with range (min–max). Categorical variables are provided as absolute numbers and in percent. In the main analysis, re-resection rate (yes/no) was analyzed using generalized mixed models for binary outcomes (PROC GLIMMIX). Since several different resections and re-resections were required in some patients, the repeated measure structure of the data was taken into account in these models. Therefore, a variance components covariance structure was used. Odds ratios (OR) and 95% confidence intervals (95% CI) were calculated. For the primary analysis, the impact of patient’s sex, surgeon’s sex, and surgeon–patient sex concordance and discordance (female/female, female/male, male/female, male/male) was separately analyzed. The impact of secondary variables (age, surgery duration, safety margin, diagnosis (BCC, EC, SCC, MIS, MM), location (nasal, frontotemporal, orbital, buccal, oral), wound closure (direct wound closure, healing by secondary intention, others), type of BCC (not documented, mixed, basosquamous, micronodular, infiltrative, morpheaform, superficial, nodular), and ulceration (yes/no)) were analyzed as univariates. All variables showing a *p*-value of *p* < 0.05 in univariate analyses were included in the multivariate analysis. A *p*-value < 0.05 was considered statistically significant. Statistical analyses were performed using SAS 9.4 (SAS Institute Inc., Cary, NC, USA).

## 3. Results

### 3.1. Patients

We included 469 patients who underwent the excision of an FSC with 1 to 6 resections, resulting in a total number of 819 surgeries. The same analysis was conducted for primary FSC excisions only. However, there were no differences concerning which endpoints showed a significant association with the re-resection rate; therefore, these results are not presented. Ranging from 21 to 96 years, the mean age of the included patients at first resection was 69 ± 15 years. The sex ratio of the patients was almost equally distributed, with 213 surgeries (45.4%) being performed on men and 256 (54.6%) on women. By far the most common diagnosis of the 819 resections was BCC (*n* = 487, 59.5%), followed by cSCC (*n* = 208, 25.4%), MM (*n* = 80, 9.8%), MIS (*n* = 32, 3.9%), and EC (*n* = 12, 1.5%). Concerning the location, the majority of FSCs were located in the nasal area (*n* = 281, 34.3%), followed by the frontotemporal (*n* = 171, 20.9%), buccal (*n* = 169, 20.6%), orbital (*n* = 110, 13.6%), and oral (*n* = 88, 10.7%) area. Amongst a total of 487 BCCs, the following subtypes were identified according to the histopathological report: nodular (*n* = 179, 36.8%), infiltrative (*n* = 51, 10.5%), superficial (*n* = 47, 9.7%), mixed (*n* = 22, 4.5%), morpheaform (*n* = 17, 2.1%), micronodular (*n* = 13, 2.7%), basosquamous (*n* = 6, 1.2%), and not documented (*n* = 152, 31.2%).

### 3.2. Sex-Specific Reoperation Rate

An imbalance was detected in the sex ratio of the surgeons: whereas most surgeries (*n* = 619, 75.6%) were performed by male surgeons, the minority (*n* = 200, 24.4%) were performed by female surgeons. Amongst 819 surgeries performed by a total of 43 surgeons, in 397 cases (48.5%) the patients were sex concordant with their surgeon (male surgeon with male patient, *n* = 290, 35.4%; female surgeon with female patient, *n* = 107, 13.1%), while 422 (51.5%) were sex discordant (male surgeon with female patient, *n* = 329, 40.2%; female surgeon with male patient, *n* = 93, 11.4%). In a total of 166 patients (35.4%), at least one re-resection was performed later; among them, 84 (50.6%) re-resections were performed on female patients and 82 (49.4%) on male patients, while in 303 patients (64.6%) no secondary surgery was needed.

The univariate analysis revealed no significant differences concerning the re-resection rate in dependency of the sex of the surgeon (*p* = 0.653; OR: 1.09, 95% CI: 0.76–1.56) and the patient (*p* = 0.323; OR: 0.85, 95% CI: 0.62–1.17), as well as surgeon–patient sex concordance and discordance (*p* = 0.696). The association between sex-specific factors and the re-resection rate of FSC is presented in Table 1.

### 3.3. Secondary Endpoints

Several significant associations between secondary endpoints and the re-resection rate were revealed in the univariate analysis (Table 2).

Healing by secondary intention (*p* ≤ 0.001; OR: 3.19; 95% CI: 2.10–4.83) showed a significant direct association with the probability for secondary surgery when taking direct wound closure as a reference. However, if more complex wound closures or reconstructive procedures had been performed, the likelihood of a further re-resection was significantly decreased (*p* = 0.011; OR: 0.60; 95% CI: 0.40–0.89). Additionally, a longer surgery duration (*n* = 365) was associated with a trend towards an increased likelihood of secondary surgery (*p* = 0.055; OR: 1.007; 95% CI: 1.000–1.014). A bigger safety margin (*n* = 343) showed a significantly decreased re-resection rate (*p* = 0.011; OR: 0.92; 95% CI: 0.86–0.98). Furthermore, the univariate analysis revealed a significantly decreased probability of a further re-resection of FSCs located in the frontotemporal (*p* = 0.009; OR: 0.57; 95% CI: 0.38–0.87), orbital (*p* = 0.023; OR: 0.56; 95% CI: 0.34–0.92), and buccal (*p* < 0.001; OR: 0.33; 95% CI: 0.20–0.53) areas, compared to FSCs, which were located in the nasal area. However, no significant association was found between the nasal and mouth area (*p* = 0.819; OR: 0.94; 95% CI: 0.57–1.56), suggesting a similar re-resection rate. 

The following observations concerning the association between the diagnosis and the re-resection rate were revealed: cSCC (*p* = 0.014; OR: 0.61; 95% CI: 0.41–0.90) and MM (*p* = 0.048; OR: 0.66; 95% CI: 0.30–0.99) were associated with a significantly decreased likelihood, while EC (*p* = 0.008; OR: 6.80; 95% CI: 1.65–27.97) had a significantly increased likelihood of a further re-resection, compared to BCC. No significant difference in re-resection rate was observed in MIS, compared to BCC. The age of the patient (*p* = 0.106), BCC subtype (*p* = 0.327), and the level of training of the surgeon(s) (*p* = 0.807) showed no significant association with the re-resection rate in the univariate analysis.

The multivariate analysis revealed that healing by secondary intention was the only independent predictor for a subsequent re-resection, with significant differences in re-resection rate in comparison to direct wound closure (*p* < 0.001; OR: 4.28; 95% CI: 1.94–9.45) (Table 3).

## 4. Discussion

In this single-center, retrospective study, we found consistent evidence that sex-specific factors had no significant association with the re-resection rate in the surgical treatment of FSC. However, healing by secondary intention, surgery duration, resection safety margin, cancer location, and the type of skin cancer showed a significant association with the re-resection rate.

According to our knowledge, this is the first study focusing on sex-specific differences, as well as the impact of cancer- and operation-specific factors, in the resection of FSC. The majority of sex studies in the field of medicine focus on clinical practice, such as physician–patient relationship and communication [17,18,19]. In fact, very few studies have been conducted in the field of surgery—amongst them, a population-based cohort study, which was published in JAMA *Surgery* in February 2022. The study examined the impact of surgeon–patient sex concordance on the surgical outcome of patients undergoing common elective and emergent procedures such as hip replacement and carpal tunnel release. The main finding was that surgeon–patient sex discordance was associated with an increased likelihood of adverse postoperative outcomes, which were defined as complications, readmission, and death [12].

In contrast to that, our study showed that surgeon–patient sex discordance, as well as other sex-specific factors, had no impact on the re-resection rate in the surgical treatment of FSC. However, the likelihood of a subsequent re-resection was associated with other operation- and cancer-specific factors. In our multivariate analysis, healing by secondary intention showed a strong association with an increased likelihood of a subsequent re-resection in comparison to direct wound closure. According to the authors’ knowledge and expertise, this outcome might be due to the intraoperative assessment of the surgeon suspecting a possible incomplete excision because of the high complexity of the surgery or a tricky location such as the nose and mouth area. In fact, this suggests the benefits of a two-stage surgical procedure, including standard excision with delayed reconstruction of the operation site until complete excision is confirmed by histopathological evaluation, especially in patients with FSCs characterized by an increased risk of incomplete excision. The literature suggests that further surgical interventions, in order to prevent or treat recurrence after incomplete skin cancer excisions, are associated with an increase in cost, time, surgical complexity, and patient morbidity [15]. These outcomes emphasize the recommendations of care for non-melanoma skin cancers according to the *Journal of the American Academy of Dermatology*. The guidelines suggest favoring a two-stage technique, including healing by secondary intention for the surgical treatment of high-risk skin cancers [20,21].

Further analyses revealed a significant univariate association between a longer duration of surgery and an increased likelihood of re-resection. According to our knowledge, this outcome might be due to the higher complexity of the surgery, as well as intraoperative complications, which are associated with a longer surgery duration. Therefore, the duration of surgery is to be considered as a sequela of the surgery complexity, which could explain the higher re-resection rate and not the surgery duration itself.

Also, a bigger safety margin showed a significantly decreased re-resection rate. We already suspected this outcome, since the probability of an in sano resection increases with the amount of tissue, which is removed during surgery. These results also show the meaningfulness of the definition of different safety margins in dependency of the type of skin cancer, as well as other operation-specific factors such as skin cancer location, which are already implemented in general practice and should be strictly followed by surgeons.

Furthermore, the type of skin cancer showed a significant association with the probability of a subsequent re-resection; whereas cSCC and MM were associated with a significantly decreased likelihood, EC had a significantly increased likelihood of a further re-resection, compared to BCC. According to the authors’ knowledge and expertise, the higher re-resection rate of BCC might be due to the cancer’s entity by itself, which is usually referred to as semi-malign because of its low metastatic potential (<0.01%) and mortality rate (0.12 per 100,000) [3]. Therefore, the surgeon might underestimate the extension of the lesion and take a less invasive approach by using a smaller safety margin or a rather superficial excision, causing a relatively high re-resection rate. In comparison to that, a more invasive approach might be taken when operating on patients with cSCC, and especially MM, according to the increased metastatic potential and mortality rate. Also, patients with metastatic skin cancer or those, who underwent surgery in a palliative setting, were excluded from the study, which might further have an impact on the presented results. Regarding the fact that EC showed a significantly increased re-resection rate, we believe this might be due to this type of cancer being extremely rare, accounting for only 0.005–0.01% of all malignant cutaneous neoplasms. Therefore, surgeons might not be as experienced with the surgical treatment in comparison to the well-known types of skin cancers. This fact, in combination with the aggressive nature and relatively high local recurrence rate of 20%, might be possible explanations for the higher re-resection rate in EC [22]. However, due to the limited amount of EC cases (*n* = 12), future studies are necessary to assess the re-resection rate of EC, as well as factors that have a significant impact, in order to optimize the surgical treatment.

Moreover, the location of the cancer had a significant univariate association with the re-resection rate, whereas FSCs located in the nasal and mouth area showed an increased likelihood of a subsequent re-resection. The nose and mouth are part of area H, also referred to as the “mask area” of the face [15]. According to National Comprehensive Cancer Network Guidelines, BCCs and cSCCs in these areas are classified as high risk for recurrence independently of other cancer characteristics; therefore, more radical treatment modalities are recommended [15,20]. These facts align well with our results, which showed a higher re-resection rate in these areas, as they might be more complex to operate on due to the complexity of the anatomical structures and lack of excess tissue [16].

Obviously, our study also includes several limitations. First, this was a single-center study and therefore only represents the results of one department. Clearly, a significant amount of FSC patients underwent surgery at the Department of Dermatology and Venereology of the Medical University of Graz, Austria. Therefore, we were not able to include these patients into the study. This might explain the high re-resection rate, since more complex cases are usually referred to our plastic surgical department. Second, due to the retrospective nature, we were not able to collect all data completely in some cases. Especially with regard to the safety margin, numbers were often not documented, which explains the limited number of cases (*n* = 343) that were included in the analysis. This limitation could have been minimized by performing a prospective study. However, owing to the impossibility of blinding the sex of the surgeons and patients, this is not a major limitation. Third, besides the above-mentioned factors, other potentially important factors such as social determinants, communication styles, and uncontrollable biases might influence the surgeon–patient interaction and were not able to be captured in this study. An interesting analysis would have been if the size of the skin cancer has an impact on the re-resection rate; however, due to the lack of documentation and discrepancies between the operation and histopathological report, we did not include this endpoint in our study. Also, the re-resection rate of re-excision surgeries was not evaluated; however, according to the authors opinion, this would have been biased due to the non in sano excision by itself. Moreover, the study only refers to the biological sex of the surgeons and patients and we were not able to assess gender, which might meaningfully influence physician–patient interactions. Finally, race and ethnicity had a significant influence on the physician–patient relationship according to the study published by Cooper-Patrick et al., the impact of which was not assessed in this study [23]. Nevertheless, as most patients treated at our plastic surgical department are White, assessing the impact of race and ethnicity would not have led to meaningful results.

## 5. Conclusions

In conclusion, this single-center retrospective study demonstrated no significant association between surgeon–patient sex concordance, as well as other sex-specific factors, and the re-resection rate in the surgical treatment of FSC. However, further analyses revealed other operation- and cancer-associated factors, which have an impact on the likelihood of a subsequent re-resection. Healing by secondary intention showed a significantly increased re-resection rate in comparison to direct wound closure, which suggests the benefits of a two-stage surgical procedure. A longer surgery duration was associated with an increased probability of second surgery, which might be due to a higher surgery complexity. A bigger safety margin showed a decreased re-resection rate, which shows the meaningfulness of the definition of different safety margins in dependency of cancer- operation- and patient-specific factors. EC showed a higher re-resection rate in comparison to other FSCs, which indicates the necessity to place more emphasis on the surgical treatment of rare types of skin cancers. FSCs located in the nose and mouth area were associated with an increased likelihood of secondary surgery, which aligns well with the fact that these areas are characterized as having a high-risk of recurrence. The awareness of these factors, which have an impact on the re-resection rate, and the knowledge of the interaction between them might be used in the future to find an algorithm to provide a personalized treatment to patients suffering from FSCs. The ultimate goal should be to limit the number of surgeries needed and therefore reduce the strain on the patient and the burden on the health care system. Finally, to fully understand the risk of secondary surgery in dependence of patient-, operation-, and cancer-specific factors on physician–patient relationships, further studies including multiple centers are warranted to also detect regional, ethnic, and racial differences.

## Figures and Tables

**Figure 1 jpm-13-01193-f001:**
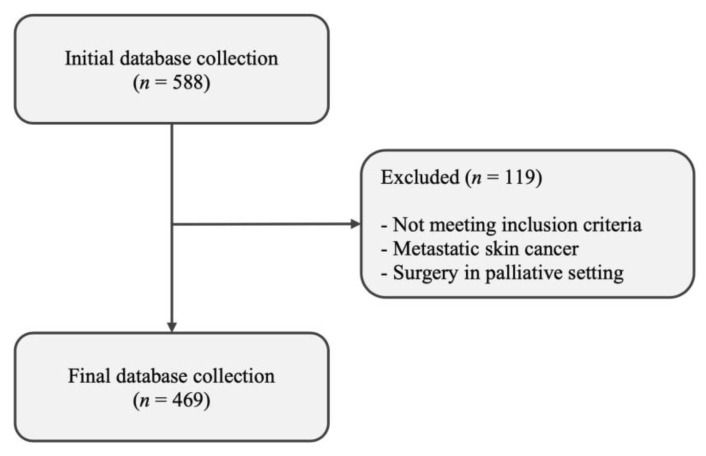
Patient cohort flow diagram.

**Figure 2 jpm-13-01193-f002:**
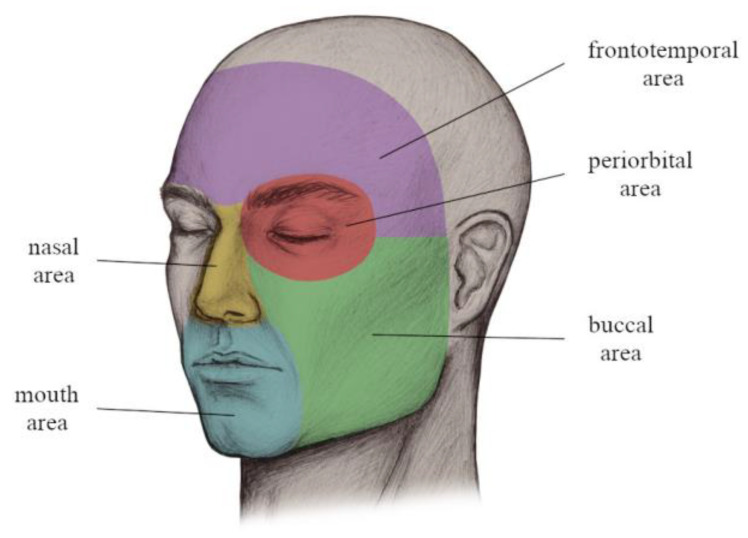
Facial skin cancer locations.

**Table 1 jpm-13-01193-t001:** Univariate analysis of sex-specific differences in the resection of facial skin cancer (*n* = 469, surgeries: 819). Pat: patient; OP: surgeon; OP2: second surgeon/assistant; F: female; M: male.

	n	OR (95% CI)	*p*-Value
OP sex	819	1.09 (0.76–1.56)	0.653
OP2 sex	463	0.83 (0.52–1.31)	0.410
Pat/OP sex	819		0.727
Pat: M/OP: M			Ref
Pat: M/OP: W		0.98 (0.58–1.67)	0.944
Pat: W/OP: M		0.81 (0.56–1.17)	0.265
Pat: W/OP: W		0.98 (0.58–1.67)	0.899
Sex concordance	819	0.86 (0.62–1.17)	0.333

**Table 2 jpm-13-01193-t002:** Univariate analysis of secondary endpoints (*n* = 469, surgeries: 819).

	n	OR (95% CI)	*p*-Value
Age	819	0.99 (0.98–1.00)	0.106
Sex	819	0.85 (0.62–1.17)	0.323
Surgery duration (min) ^1^	365	1.007 (1.000–1.014)	0.055
Safety margin (mm)	343	0.92 (0.86–0.98)	0.011
Diagnose	819		0.005
BCC			Ref
EC		6.80 (1.65–27.97)	0.008
cSCC		0.61 (0.41–0.90)	0.014
MIS		0.84 (0.37–1.92)	0.685
MM		0.55 (0.30–0.99)	0.048
Location	819		<0.001
Nasal			Ref
Frontotemporal		0.57 (0.38–0.87)	0.009
Orbital		0.56 (0.34–0.92)	0.023
Buccal		0.33 (0.20–0.53)	<0.001
Oral		0.94 (0.57–1.56)	0.819
Wound closure	819		<0.001
Direct wound closure			Ref
Others		0.60 (0.40–0.89)	0.011
Healing by secondary intention		3.19 (2.10–4.83)	<0.001
BCC subtype	487		
Not documented			
Mixed		1.13 (0.70–1.83)	0.612
Basosquamous		1.11 (0.54–2.28)	0.774
Micronodular		1.83 (0.66–5.13)	0.248
Infiltrative		1.99 (1.03–3.84)	0.041
Morpheaform		3.06 (0.97–9.62)	0.056
Superficial		1.31 (0.23–7.42)	0.760
Nodular		1.50 (0.59–3.83)	0.400
Ulceration	487	1.09 (0.70–1.69)	0.701
Type of anesthesia	819	1.02 (0.74–1.42)	0.901

^1^ only cases with direct wound closure or healing by secondary intention were included in the analysis.

**Table 3 jpm-13-01193-t003:** Multivariate analysis of secondary endpoints (*n* = 95, surgeries = 345).

Multivariate Solutions for Fixed Effects			
Effect	Group	OR (95% CI)	*p*-Value
Surgery duration (min)		1.016 (0.996–1.036)	0.111
Safety margin (mm)		0.981 (0.852–1.130)	0.793
Diagnosis			0.124
	BCC		Ref
	MM	0.131 (0.020–0.86)	0.034
	MIS	0.271 (0.028–2.571)	0.255
	cSCC	1.367 (0.381–4.909)	0.632
	EC	>100 (<0.001–>100)	0.726
Location			0.583
	Nasal		Ref
	Frontotemporal	1.414 (0.330–6.064)	0.641
	Orbital	1.496 (0.366–6.122)	0.575
	Buccal	0.714 (0.208–2.446)	0.592
	Oral	0.456 (0.102–2.033)	0.303
Wound closure	Direct wound closure	Ref	0.002
	Healing by secondary intention	4.43 (1.70–11.54)	

## Data Availability

The data presented in this study are available on request from the corresponding author.

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
