# Peer review of "Do Sex-Specific Factors Influence the Surgical Treatment of Facial Skin Cancer?"

_jpm, 2023, doi:10.3390/jpm13081193_

Round 1

Reviewer 1 Report

I was surprised by the small number of FSC surgeries performed in the 16 years described,  and also from the high percentage of MM, and of the relatively  overall high rate of re-resection
other than that all makes sense.

I was wondering whether the significantly higher percentage of re-resection in the patient group who's wound were left to heal by secondary intention was a result of the fact that these procedures are often performed as either a diagnostic incisional biopsy, Or as a tangential excision which is done either as a diagnostic procedure or as a procedure performed on a wrongfully clinically evaluated  as a benign skin lesion. Please elaborate on this group , the indication' the clinical diagnosis, the  surgical technique, the concordance of pre and post operative surgical diagnosis , the  reason leading for re- excision.

I was also intrigued by the fact that the patient group has patients who required up to 11 re-excisions. Do you not have the option for real time surgical margin control in order to avoid so many re-resections? Have you included all 10 re-resections in your analysis? Should it be designed like this ?

I would suggest considering   to also  perform another analysis of only "first time surgeries " and evaluate the same parameters on this group .

I would also suggest to evaluate the rate of re-excision' in a re-excision surgery.

Reviewer 2 Report

Thank you for you study looking at the influence of gender on facial skin cancer excision. It is somewhat reassuring to see that gender does not have a significant impact on incomplete excision rates.

Was lesion size looked at at all? Also, was there any difference in wound closure methods between surgeons (eg one surgeon only used secondary intention healing and had a very high re operation rate?)
